# In Vivo Evaluation of the Efficacy of a Nisin–Biogel as a New Approach for Canine Periodontal Disease Control

**DOI:** 10.3390/pharmaceutics14122716

**Published:** 2022-12-04

**Authors:** Eva Cunha, Luís Miguel Carreira, Telmo Nunes, Marta Videira, Luís Tavares, Ana Salomé Veiga, Manuela Oliveira

**Affiliations:** 1CIISA—Centro de Investigação Interdisciplinar em Sanidade Animal, Faculdade de Medicina Veterinária, Universidade de Lisboa, Av. da Universidade Técnica, 1300-477 Lisbon, Portugal; 2Laboratório Associado para Ciência Animal e Veterinária (AL4AnimalS), 1300-477 Lisbon, Portugal; 3Casa dos Animais de Lisboa, Estrada da Pimenteira, 1300-459 Lisbon, Portugal; 4Instituto de Medicina Molecular, Faculdade de Medicina, Universidade de Lisboa, Avenida Professor Egas Moniz, 1649-028 Lisbon, Portugal

**Keywords:** nisin–biogel, periodontal disease, dogs, clinical trial

## Abstract

Periodontal disease (PD) is a common oral disease in dogs. Recent in vitro research revealed that nisin–biogel is a promising compound for canine PD control. In this work, a clinical trial was developed to assess the in vivo efficacy of nisin–biogel in dogs by determining the dental plaque index (DPI), gingivitis index (GI), and periodontal pocket depth (PPD) after dental administration. The biogel’s influence on aerobic bacteria counts was also evaluated, as well as its acceptance/adverse effects in dogs. Twenty animals were allocated to one of two groups: a treatment group (TG) subjected to a dental topical application of nisin–biogel for 90 days and a control group (CG) with no treatment. Besides daily monitoring, on day 1 (T0) and at the end of the assay (T90), animals were subjected to blood analysis, periodontal evaluation, dental plaque sampling, scaling, and polishing. Statistical analysis with mixed models showed a significant reduction in mean PPD (estimate = −0.371, *p*-value < 0.001) and DPI (estimate = −0.146, *p*-value < 0.05) in the TG animals at T90. A reduction in the GI (estimate = −0.056, *p*-value > 0.05) was also observed but with no statistical significance. No influence on total bacterial counts was observed, and no adverse effects were detected. The nisin–biogel was revealed to be a promising compound for canine PD control.

## 1. Introduction

Periodontal disease (PD) is one of the most prevalent inflammatory diseases in dogs [1]. Affecting over 80% of animals over two years old, PD is initiated by the formation of a polymicrobial biofilm on the tooth surface; this is also known as dental plaque and is responsible for a subsequent local host inflammatory reaction [1,2]. PD can evolve from a reversible stage of gingivitis to an irreversible stage of periodontitis [3]. The periodontal damage can be assessed by evaluating gingivitis, furcation, and mobility indices or stages and by measuring the periodontal pocket depth and the clinical attachment level [4]. According to the severity of the periodontium damage, four stages can be considered in PD classification. Stage 1 includes animals with only gingivitis; stage 2 includes animals with early periodontitis with less than 25% of attachment loss and/or stage 1 furcation; stage 3 comprises animals with moderate periodontitis, revealing an attachment loss of 25 to 50% and/or stage 2 furcation involvement; and the final stage, stage 4, includes dogs with advanced periodontitis, more than 50% of attachment loss and/or stage 3 furcation [4,5]. Besides damaging the periodontium, the persistent contact of dental plaque bacteria with periodontal structures facilitates their migration to the bloodstream and, consequently, the appearance of PD-related systemic consequences in distant organs [6,7,8,9].

Several strategies can be used for PD control. Professional removal of the dental plaque along with the application of measures that inhibit dental plaque formation, such as daily toothbrushing (the gold standard method), application of chemical or natural anti-plaque compounds, administration of dental treats and biscuits, or a specific dental diet, are useful for PD control [4]. However, when periodontitis is present, apart from the previously mentioned measures and according to the case severity, PD treatment may include specific surgical approaches, the use of host modulation drugs, and even antimicrobial therapy [5].

Recently, an innovative approach based on the application of the antimicrobial peptide nisin incorporated in a delivery system formed by guar gum gel (nisin–biogel) has shown inhibitory and eradication abilities against pre-formed biofilms composed by PD enterococci, as well as towards canine dental plaque polymicrobial biofilms, in vitro [10,11]. Previous studies have also revealed that nisin–biogel activity is maintained in the presence of canine saliva and over a long-term storage period at distinct temperatures. Furthermore, nisin–biogel has shown an absence of cytotoxicity up to 200 µg/mL towards several cell lines, reinforcing the potential of nisin–biogel as a promising compound for canine PD control [12,13].

The main goal of this in vivo study was to evaluate the influence of the long-term dental application of the nisin–biogel on the dental plaque and gingivitis indices, the periodontal pocket depth, and total oral bacterial counts in dogs through a randomized controlled clinical trial. In addition, hematological and biochemical parameters were monitored during the trial, as well as general side effects.

## 2. Materials and Methods

### 2.1. Nisin–Biogel Preparation

Nisin–biogel was prepared as described elsewhere [10,11,13,14]. According to previous studies, a final concentration of 200 µg/mL was selected to be used in the clinical trial [10,11,12,13]. After preparation, the nisin–biogel was stored at 4 °C until further use.

### 2.2. Dog Selection

Animals were selected according to the Veterinary Oral Health Council (VOHC) guidelines for trials testing compounds for PD prevention. The dogs were from an official animal rescue institution (“Casa dos Animais de Lisboa”), and all experimental procedures were approved by the Ethical Committee for Research and Teaching (CEIE) of the Faculty of Veterinary Medicine, University of Lisbon, Portugal (N/Ref 014/2020).

The inclusion criteria were as follows: healthy dogs over 2 years old without severe PD and with no history of antimicrobial therapy in the last month. All animals were submitted to a clinical examination, oral handling, and complete blood analysis (hemogram and measurement of urea, creatinine, alanine aminotransferase, alkaline phosphatase, glucose, albumin, and total blood proteins) to detect any deviations that would prevent their inclusion in the study.

### 2.3. Clinical Trial

A total of twenty dogs were selected and submitted to a complete periodontal evaluation, dental plaque sampling (for bacterial total counts), scaling, and dental polishing (timepoint 0). The complete removal of dental plaque and calculus was checked using a disclosure solution (GC Tri Plaque ID Gel^®^, Tokyo, Japan). All procedures were performed under general anesthesia using acepromazine (0.01 mg/kg, IM), propofol (2 mg/kg, IV), and isoflurane. Intraoperative meloxicam (0.2 mg/Kg, SC) and amoxicillin/clavulanate (8.75 mg/kg, SC) were administered to all animals [15]. After that, each dog was randomly allocated to one of two groups: a treatment group (TG, N = 10) or a control group (CG, N = 10). Animals in the treatment group were submitted to a topical dental application of the nisin–biogel (200 µg/mL) every 48 h, as mentioned in Table 1. The control group was composed of dogs that were not submitted to any treatment. Animals were kept in the trial for 90 days. At the end of the clinical trial (T90), all animals were subjected to a new complete blood analysis, complete periodontal evaluation, dental plaque sampling, scaling, and dental polishing.

In addition, two intermediate dental plaque evaluations of all animals were performed on days 30 and 60. Intermediate evaluations included an awake dental plaque evaluation using a disclosure solution (GC Tri Plaque ID Gel^®^, Tokyo, Japan) that dyed the dental plaque according to its accumulation. Then, a photographic register of the vestibular margins was performed on all animals. After that, the dental plaque coverage of 4 teeth (2 canines and 2 premolars) of each dog was evaluated using the IMAGEJ^®^ program. The percentage of dental plaque coverage was determined after measuring the total tooth vestibular margin and the area of the dyed dental plaque.

Animals in both groups were fed the same dry food and housed in the same building. All animals were observed daily to detect any general side effects, such as changes in feeding habits and behavior, prostration, vomiting, or diarrhea.

### 2.4. Periodontal Evaluation

The periodontal examination was blinded and performed by a trained veterinarian. The clinician determined the dental plaque and gingivitis scoring and performed six measures of the distance between the gingival margin and the bottom of the periodontal pocket (periodontal pocket depth—PPD) of each tooth (three measures were performed in the vestibular face and the other three in the palatine face) [16,17]. Every tooth in each animal was evaluated. The indexes used are presented in Table 2 and Table 3.

### 2.5. Dental Plaque Sample Processing

A dental plaque sample was collected from all dogs on day 0 and day 90 using a swab (AMIES, VWR, Amadora, Portugal), which was applied to the entire dental surface. Swabs were transported to the Laboratory of Microbiology and Immunology, Faculty of Veterinary Medicine, University of Lisbon, and processed for total bacterial quantification, according to Belo et al. (2018) [18]. Briefly, the collected swabs were placed in test tubes with 1 mL of sterile saline and vortexed, and the resulting suspension was diluted (10^−1^ to 10^−8^). From each dilution, 100 μL were collected and inoculated on Brain Heart Infusion agar (VWR, Amadora, Portugal) and incubated at 37 °C for 48 h. Afterward, bacterial quantification was performed by determining the colony-forming units [18].

### 2.6. Statistical Analysis

Statistical analysis of the data was carried out using RStudio^®^ software version 1.1.383 (Boston, MA, USA) and Microsoft Excel 2016^®^ (Redmond, WA, USA). Variables were evaluated by plotting the data into a histogram to confirm that they followed a normal distribution. Linear mixed models were used to evaluate differences in the dependent variables: mean PPD, GI, and DPI. In the mixed model, the variables group, timepoint, and tooth type (superior or inferior, incisive, canine, pre-molar, and molar) were defined as fixed effects, and the variables animal, weight, and tooth number were considered random effects. The interaction between fixed effects was investigated, and the Akaike Information Criterion was used to select the model.

Quantitative variables were expressed as mean values ± standard deviation. A confidence interval of 95% was considered in this study, with a *p*-value ≤ 0.05 indicating statistical significance.

## 3. Results

A total of 20 dogs were selected to participate in this clinical trial. The animals’ gender distribution and mean age and weight are presented in Table 4. The results of the hemogram and biochemical parameters of all animals included in the trial agreed with the reference values established for healthy dogs at both timepoints (0 and 90) (Appendix A).

### 3.1. Periodontal Evaluation

Complete periodontal evaluations of every tooth of each dog were performed at timepoints 0 and 90. The mean results for each dog are presented in Table 5. At timepoint 0, after periodontal assessment, all dogs were submitted to scaling and dental polishing before proceeding to the clinical trial.

Statistical analysis with mixed models showed a significant reduction in mean PPD (estimate = −0.371, *p*-value < 0.001) for all tooth types evaluated (incisive, canine, molar, and pre-molar) in the animals in the treatment group at T90. For the DPI, a statistically significant reduction (estimate = −0.145, *p*-value < 0.05) was observed in the animals in the treatment group at T90, with a marked reduction (*p*-value < 0.001) in incisive teeth. In addition, a reduction in the GI was observed in the animals in the treatment group at T90, but it was not statistically significant (estimate = −0.056, *p*-value > 0.05). For the GI, it was possible to observe a significant reduction (*p*-value < 0.05) in the mandibular incisive teeth and the maxillary molars and pre-molars.

Intermediate evaluation of dental plaque coverage performed on days 30 and 60 using a disclosure solution revealed that the animals in the treatment group presented a mean dental plaque coverage of 24.58% and 33.08%, respectively. On the other hand, the dogs in the control group presented a mean dental plaque coverage of 42.37% on day 30 and 51.61% on day 60.

The results of the intermediate dental plaque coverage obtained by IMAGEJ are presented in Table 6.

### 3.2. Dental Plaque Sample Processing

Total oral aerobic bacteria counts were performed on days 0 and 90 using the dental plaque swab samples obtained from each animal. At T0, the samples revealed a mean total count of 8.6 × 10^7^ CFU/mL, with those from the treatment group presenting 6.9 × 10^7^ CFU/mL and those from the control group presenting 1 × 10^8^ CFU/mL. At T90, the mean total counts were 7.4 × 10^7^ CFU/mL, with the samples from the treatment group presenting 8.3 × 10^7^ CFU/mL and those from the control group presenting 6.6 × 10^7^ CFU/mL.

## 4. Discussion

Periodontal disease (PD) is well established as one of the most common oral inflammatory diseases in dogs, with 80% of these animals presenting some degree of PD by two years of age [2,19]. Its high prevalence, along with its potential local and systemic consequences, reinforces the need to improve PD control measures [6,7,8,9]. Nisin–biogel has been shown to be a promising compound against canine oral biofilms in vitro [10,11], with the ability to act on periodontopathogens present in the canine oral microbiome [20]. In this study, an in vivo clinical trial was performed to assess the efficacy of nisin–biogel dental topical application in dogs. A total of twenty animals were included in the trial after health parameter assessment through blood analysis to confirm the animals’ health. During the study, all animals were observed daily to identify any adverse clinical signs or behavior deviations. Previous reports have described an absence of cytotoxicity of nisin at 200 µg/mL [12,13,21,22], which was confirmed by our study, in which none of the animals showed side effects during the 3-month trial. Moreover, a blood analysis was performed in all dogs at the end of the study, and it was observed that none presented deviations in the evaluated parameters.

In addition, oral samples were collected from all animals at T0 and T90 to evaluate the influence of the nisin–biogel on oral aerobic bacteria through total bacterial counts. No reduction in total bacteria counts was observed in the samples collected after the application of the nisin–biogel for 3 months, which is in agreement with previous reports that have suggested resilient behavior of dental plaque and oral microbiota after professional dental cleaning [15,23]. In addition, Cunha and collaborators (2021), who studied the influence of the dental application of the nisin–biogel in the dynamics of the oral microbiome of dogs, detected an increase in bacterial diversity after one week of application of this compound [20]. Considering these studies, the oral bacteria population seems to suffer a rearrangement after antimicrobial compound administration or physical aggression, maintaining microbiota dynamics and relative concentrations [20,23].

One of the main goals of this study was to evaluate the in vivo efficacy of nisin–biogel for PD control using periodontal measures, such as DPI, GI, and PPD, and it was observed that the nisin–biogel had the ability to reduce dental plaque accumulation. This reduction was particularly evident after 1 month of dental topical application, which resulted in 24.58% dental plaque coverage in the treatment group and 42.37% in the control group. Similar results were obtained by Howell et al. (1993), who applied a nisin mouth rinse in dogs for 88 days. In that study, researchers observed a 34–38% reduction in the dental plaque index in the animals submitted to nisin application [24]. Dental plaque is one of the key factors in PD onset, being responsible for the initial aggression to the periodontium [1,4]. Considering this, dental plaque reduction is an essential step for PD control [5]. At the end of our trial, a significant reduction in DPI was observed in the animals in the treatment group in comparison with those in the control group, reinforcing the potential of the nisin–biogel for PD management. This reduction may be caused by the direct action of nisin–biogel on the inner layers of the biofilm. Previous reports have shown that this compound can not only inhibit biofilm formation but also eradicate mono and polymicrobial oral biofilms [10,11]. It has been suggested that nisin’s biochemical structure and mechanism of action contribute to its high activity against biofilms [25]. Specifically, nisin can penetrate these structures without being neutralized or bound by the biofilm cells or matrix and is able to target the extracellular polymeric substances of the surrounding matrix [26,27].

The other periodontal indicator evaluated was the gingivitis index. PD can be classified into four different stages, the first of which is characterized by gingivitis [4]. This is a reversible stage in which most periodontal structures are undamaged, with animals showing inflammation and swelling of the gingiva with or without bleeding upon probing [1,4]. At the end of our clinical trial, it was possible to observe that the application of the nisin–biogel induced a slight reduction in GI. Howell and collaborators (1993) also observed a reduction in GI after nisin oral application for 88 days, which may be related to the potential immunomodulatory effect of this antimicrobial peptide [24]. In fact, besides having antimicrobial activity against several periodontopathogens, nisin presents immunomodulatory and hound healing abilities [12,28]. It acts by binding to Lipid II, present in the bacterial cell membrane, and by interfering with cell wall biosynthesis, leading to bacterial death [10]. In addition, it has been shown that nisin promotes a reduction in pro-inflammatory cytokines, which contributes to PD progression, and presents some effects on immune cells [28,29,30]. In our trial, a significant PPD decrease was observed in the animals from the treatment group at T90, which may be directly related to dental plaque and gingivitis reduction. It is known that gingiva swelling can increase pocket depth, so the reduction of gingivitis after nisin–biogel therapy may have decreased local inflammation, leading to gingival shrinkage and reducing PPD [4].

It is important to note that the animals in the control group also exhibited a slight reduction in GI and PPD at T90, with no statistical significance, which was potentially caused by the scaling and dental polishing performed at the beginning of the study. In fact, dental scaling, polishing, irrigation, and home dental care are the standard procedures for the treatment of PD stage one cases (gingivitis), so a reduction in this periodontal parameter was expected. Nevertheless, the use of nisin–biogel promoted a higher and statistically significant decrease of the DPI and PPD and a reduction in the GI compared with the results from the control group. Moreover, nisin–biogel was well accepted by the animals, showing no adverse effects over a 3-month period of application.

This study has some limitations, such as the number of animals included, and a larger in vivo study would help to validate the nisin–biogel as a potential future compound to be applied by clinicians. Yet, the fact that it was necessary to use dogs with similar housing conditions, which needed to be maintained over a long period of time, prevented the inclusion of more animals in the trial. In addition, although the oral cavity is a complex and diverse environment [11,20], in our study we focused on supragingival dental plaque and, consequently, on the aerobic microbiota. However, anaerobic bacteria are also present in dental plaque and should be evaluated in a future trial [11,20]. Finally, as PD establishment is a multifactorial process [31,32] influenced by the oral microbiome, the host immune system and the local oral environment [1,4,33], in upcoming studies the influence of the nisin–biogel in the oral immune-inflammatory response could be a relevant issue to explore [33].

Considering the high prevalence and impact of canine PD, several other natural compounds are being studied to evaluate their potential for the control of this disease. Promising strategies include essential oils, alcoholic herbal products, and algae [34,35]. Most of these compounds have already been evaluated in vitro regarding their antimicrobial abilities. Moreover, in vivo studies have already shown the efficacy of oral products, including *Calendula officinalis* or the brown alga *Ascophyllum nodosum*, for PD control [34,35].

The nisin–biogel seems to be a promising compound for veterinary dentistry and human medicine [31,32]. Reports have described the potential application of this compound in human odontology and other areas, including for the treatment of diabetic foot infections [36], reinforcing the versatility of the nisin–biogel use in the biomedical field.

## 5. Conclusions

Periodontal disease is an inflammatory disease that is highly prevalent in dogs. The development of effective measures to control this disease is essential. Previous in vitro studies have revealed that the nisin–biogel is a safe antibiofilm agent with the potential to be used in canine PD control. Our study aimed to evaluate the in vivo efficacy of the nisin–biogel in dogs. Nisin–biogel showed the ability to reduce the DPI, GI, and PPD without any adverse effects on the animals in the study. These results suggest that the topical dental application of nisin–biogel may be used as an adjuvant measure for the prevention and control of canine PD. In the future, a large in vivo trial would be useful to fully validate this compound for commercial use. In addition, considering the high similarity between dog and human PD [31,32], nisin–biogel could also be a valuable compound to use in human dentistry.

## Figures and Tables

**Table 1 pharmaceutics-14-02716-t001:** Posology of the nisin–biogel (200 µg/mL) according to the animal’s weight.

Animal’s Weight	Volume of Nisin–Biogel
<20 kg	2 mL
20–40 kg	3 mL
>40 kg	4 mL

**Table 2 pharmaceutics-14-02716-t002:** Dental plaque index (DPI) based on Holmstrom et al. [16].

Score	Description
1	No plaque on the dental surface
2	Thin film of plaque at gingival margin detectable with probing
3	Moderate amount of plaque at gingival margin, plaque is directly visible
4	High abundance of dental plaque accumulation in the gingival margin and/or dental surface, including interdental space

**Table 3 pharmaceutics-14-02716-t003:** Gingivitis index (GI) according to the modified Talbott method [17].

Score	Description
0	Normal gingiva; no inflammation, discoloration, or bleeding
1	Mild inflammation, slight color change, mild alteration of gingival surface, no bleeding upon probing
2	Moderate inflammation, erythema, swelling, or bleeding upon probing or when pressure applied
3	Severe inflammation, severe erythema and swelling, tendency toward spontaneous hemorrhage, some ulceration

**Table 4 pharmaceutics-14-02716-t004:** Gender, age, and weight distribution by group of animals in this study.

**Global Gender Distribution** (Female/Male)	8 F/12 M
Treatment group	4 F/6 M
Control group	4 F/6 M
**Global age distribution (years)** (mean values ± SD)	5.25 ± 1.69
Treatment group (TG)	6.1 ± 1.52
Control group (CG)	4.4 ± 1.58
**Global weight distribution (Kg)** (mean values ± SD)	23.34 ± 6.30
Treatment group (TG)	25.4 ± 6.59
Control group (CG)	21.35 ± 6.04

F—Female; M—Male; SD—Standard deviation.

**Table 5 pharmaceutics-14-02716-t005:** Mean gingivitis, dental plaque, and periodontal pocket depth of each dog by timepoint.

Animal ID	Group	Mean GI		Mean DPI		Mean Palatine PPD		Mean Vestibular PPD		**Total Mean PPD**	
		T0	T90	T0	T90	T0	T90	T0	T90	T0	T90
**1**	TG	1.93	0.45	3.71	2.21	1.33	1.06	1.63	1.08	1.48	1.07
**3**	TG	2.32	0.14	2.89	1.57	1.68	1.56	2.40	1.62	2.04	1.58
**9**	TG	2.33	0.64	2.76	1.83	1.25	1.22	1.35	1.41	1.30	1.32
**10**	TG	1.83	1.59	3.40	2.10	1.99	1.36	2.25	2.10	2.12	1.67
**11**	TG	2.19	1.64	3.86	1.79	1.68	1.42	1.98	1.65	1.83	1.54
**12**	TG	1.95	0.49	2.98	2.02	1.27	1.17	1.36	1.31	1.31	1.24
**13**	TG	2.17	0.12	2.67	1.98	1.28	1.15	1.29	1.14	1.29	1.14
**14**	TG	2.63	0.00	2.83	1.19	1.75	1.08	2.12	1.05	1.93	1.06
**18**	CG	2.31	1.29	2.98	1.83	1.41	1.33	1.41	1.29	1.41	1.31
**19**	CG	2.79	0.71	3.38	2.14	1.15	1.10	1.15	1.17	1.15	1.14
**20**	CG	2.33	1.19	3.19	2.21	1.31	1.27	1.30	1.18	1.31	1.23
**22**	TG	2.21	0.71	3.24	1.26	1.45	1.17	1.57	1.29	1.51	1.23
**23**	TG	2.41	1.50	3.02	2.14	1.76	1.21	2.44	1.34	2.10	1.27
**27**	CG	1.38	0.10	2.14	1.21	1.13	1.14	1.06	1.07	1.09	1.11
**28**	CG	2.19	0.43	3.14	1.85	1.06	1.12	1.10	1.14	1.08	1.13
**31**	CG	2.05	0.10	2.88	1.36	1.21	1.15	1.20	1.07	1.20	1.11
**33**	CG	1.95	0.21	2.31	1.57	1.28	1.35	1.25	1.29	1.26	1.29
**34**	CG	2.33	1.18	3.36	2.10	2.17	2.20	2.44	1.87	2.31	2.03
**35**	CG	1.26	1.40	2.93	1.90	1.29	1.24	1.42	2.01	1.35	1.62
**37**	CG	2.24	0.12	2.68	1.57	1.04	1.22	1.20	1.25	1.12	1.23
**Mean** ± SD		**2.14** ± 0.61	**0.7** ± 0.85	**3.02** ± 0.97	**1.82** ± 0.83	**1.43** ± 0.72	**1.27** ± 0.52	**1.59** ± 0.97	**1.36** ± 0.66	**1.51** ± 0.86	**1.32** ± 0.59
**Mean T** ± SD		**2.16** ± 0.53	**0.63** ± 0.81	**3.14** ± 1	**1.84** ± 0.83 *	**1.53** ± 0.79	**1.25** ± 0.51	**1.79** ±1.08	**1.41** ± 0.69	**1.66** ± 0.96	**1.33** ± 0.61 *
**Mean C** ± SD		**2.12** ± 0.65	**0.74** ± 0.86	**2.94** ± 0.94	**1.8** ± 0.82	**1.36** ± 0.67	**1.29** ± 0.53	**1.46** ± 0.86	**1.32** ± 0.63	**1.41** ± 0.77	**1.31** ± 0.58

ID—identification; SD—standard deviation; TG—treatment group; CG—control group; T0—timepoint 0; T90—timepoint 90; *—statistical significance (*p*-value < 0.05).

**Table 6 pharmaceutics-14-02716-t006:** Mean dental plaque coverage (%) of each dog, determined after application of a disclosure solution and IMAGEJ analysis, on days 30 and 60.

Animal ID	Group	Dental Plaque Coverage (%)	
		30 Days	60 Days
**1**	TG	16.73	17.73
**3**	TG	16.82	22.52
**9**	TG	27.83	28.31
**10**	TG	26.87	32.91
**11**	TG	22.74	32.31
**12**	TG	24.54	36.99
**13**	TG	32.92	45.44
**14**	TG	13.65	28.17
**18**	CG	53.25	44.83
**19**	CG	62.16	77.03
**20**	CG	61.36	78.15
**22**	TG	38.74	39.68
**23**	TG	18.74	46.76
**27**	CG	16.88	22.65
**28**	CG	44.11	43.94
**31**	CG	40.29	36.79
**33**	CG	33.95	57.85
**34**	CG	48.11	64.12
**35**	CG	29.02	35.22
**37**	CG	34.63	55.53
**Mean** ± SD		33.38 ± 14.73	42.35 ± 17.02
**Mean T** ± SD		24.58 ± 7.89	33.08 ± 9.39
**Mean C** ± SD		42.37 ± 14.41	51.61 ± 18.24

ID—identification; SD—standard deviation; TG—treatment group; CG—control group.

## Data Availability

The datasets used and/or analyzed in the current study are available from the corresponding author upon reasonable request.

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
