# Peer review of "In Vivo Evaluation of the Efficacy of a Nisin–Biogel as a New Approach for Canine Periodontal Disease Control"

_pharmaceutics, 2022, doi:10.3390/pharmaceutics14122716_

Round 1

Reviewer 1 Report

The current paper presents a clinical trial performed in dogs, on the control of canine periodontal disease, using a nisin-based biogel.

Introduction is adequate and presents the background and motivation of this study, while providing reference to previous in vitro research using the same substance and showing the necessity of an in vivo trial. The study is original and important, since periodontal disease very is frequent in dogs and new measures to fight it are necessary.

Materials and methods are well described, the study design is appropriate and capable of yielding relevant results. The number of individuals included in the study is enough and all tests comply with the ethical norms. Statistical analysis is also relevant.

Results are well presented, the tables included are relevant and provide the reader with the necessary data to evaluate the outcome of the study.

Discussions are well conducted and compare the results to what was already included in the literature, while pointing out the originality and novelty of the study. Authors also pointed out the limitations of the study and the need for further research, using larger cohorts of animals, for commercial use of the nisin biogel. Conclusions are in accordance with the results.

Although the text is understandable, English language needs further editing, as there are still many mistakes or inappropriate phrase constructions.

The overall merit of the manuscript is high, it just needs some stylisitic improvements.  

Author Response

Dear reviewer,

We would like to thank you for the comments and the discussion points presented. We revised the manuscript aiming at improving the English language. Changes were highlighted, as you may see in the new document submitted.

Best regards,

Eva Cunha

Reviewer 2 Report

 In the current work, a clinical trial was conducted by authors to assess the in vivo efficacy of nisin-biogel in dogs, by determining dental plaque index (DPI), gingivitis index (GI) and periodontal pocket depth (PPD) after dental administration of nisin-biogel to the animals. The study was carefully designed and supported by the previously conducted studies of the group.

However, few modifications are required-

1.        Periodontitis is caused by aerobic and anaerobic bacteria both, then why authors have considered and evaluated only the aerobic colonies of oral cavity?

2.        Abstract section needs to be re-written in such a manner that results in terms of values can be discussed.

3.        Throughout the text, some words are repeated in a single line itself, therefore, complete revision of the text is needed.

Author Response

Dear Reviewer,

We would like to thank you for the comments and the discussion points presented. Considering the first question, strict anaerobes are present in the subgingival dental plaque, participating in PD progression.  We focus in the aerobic oral counts, because we aimed to understand the direct impact of our nisin-biogel in the supragingival microbiota, that is mostly composed by aerobic and facultative anaerobic bacteria. In addition, we included a paragraph in the discussion section in which we point out that anaerobic bacteria are also present in the dental plaque (see lines 89-92).

Considering the second question we changed the abstract and included the values requested; however, considering the limit of 200 words established in the journal guidelines for authors, it is difficult to include all information in the abstract.

We also revised the manuscript to improve the English language, and changes were highlighted in yellow in the new document.

Best Regards,

Eva Cunha

Reviewer 3 Report

The topic of the manuscript is to evaluate in vivo the influence of the long-term dental application of the nisin-biogel in the dental plaque and gingivitis indices, periodontal pocket depth and oral bacterial total counts, through a randomized controlled clinical trial with dogs.

The title and the abstract of the article are informative. The Introduction briefly presents the issue of periodontal disease in animals. The section "Material and Methods" precisely describes the chosen study design, however, the statistical subsection should be modified. The section "Results" should be revised from the statistical side. The Discussion is interestingly written, however, the paragraph about the study limitations and the more recent references should be supplemented. The Conclusions seem to be the "take-home" messages.

Some following points must be clarified/corrected for the further processing of this article.

Merits-related comments:

1.       The largest study limitation is the very small study group. The sample size should be increased in order to increase the reliability of statistical analyses.

2.       Did the variables correspond to the normal distribution (which should be mentioned in the methodology)? If not, the results should be presented in the form of box plots with medians and quartile stripes (and not with mean and standard errors).

3.       Why were the results of the hemogram and biochemical parameters not presented?

4.       The study limitations should be extended at the end of the Discussion.

5.       It is suggested to add more recent articles from 2020-2022 to the references in the Introduction and the Discussion.

Technical comments:

1.       The manuscript requires editorial editing, e. g. typing errors (e.g. “dept” instead of „depth”, „weigh” instead of „weight” etc.).

2.       Line “115” before the list of references should be removed.

Author Response

Dear Reviewer,

We would like to thank you for the comments and the discussion points presented. We agree that these points will improve the quality and understanding of the manuscript, and changed the manuscript accordingly.

Reference to study limitations were included in the discussion section, namely the importance of having a larger number of animals in the trial. The variables followed a normal distribution, and this information was added in lines 159-160. We added a supplementary file (supplementary file 1) with the information of the hemogram and biochemical values of the animals included in this study. New references were also added (see reference nº24, 26-28 and 34-37).

The technical comments were addressed in the new manuscript submitted, highlighted in yellow.

Best Regards,

Eva Cunha

Reviewer 4 Report

The manuscript describes the in vivo experiments on the long-term application of the nisin-biogel in the dental plaque and oral bacterial total counts etc with dogs, although the oral bacterial total counts should have been done by anaerobic cultures (instead of aerobic cultures).

[Suggestions]
Tables 5 and 6;

The referee suggests that these tables should be modified to figure-format, instead of presenting all of the data in table-format.

(The authors need to check the significant digits in the Table 5 as well as at P. 8, L. 8, and the authors need to present the standard deviations in the Table 6.)

Discussion
P. 9, L. 31, and L. 54:
"no reduction in bacteria total counts after nisin-biogel application...."
"Nisin is an antimicrobial peptide, ...."

The referee suggests that the authors need to discuss on the above points.

P. 9, L. 36:
"the nisin-biogel showed ability to reduce dental plaque accumulation."

The referee suggests that the authors should describe the reasons and the mechanisms on the above point.

P. 10, L. 72-77:
The paragraph of "In conclusion,..." could be deleted, because the manuscript includes "5. Conclusions" at P. 10, L. 78-88.

Author Response

Dear Reviewer

We would like to thank you for your comments, that we believe will contribute to improve the quality of the manuscript. Changes in Table 5 and 6 were performed as suggested, in order to include statistical information and standard deviation values. The points further discussed were mentioned in the manuscript lines 33-39, 52-59 and 69-71. We removed the final paragraph of the discussion section as suggested.

Best Regards,

Eva Cunha

Reviewer 5 Report

 The authors aimed to evaluate in vivo the influence of the long-term dental application of the nisin-biogel in the dental plaque and gingivitis indices, periodontal pocket dept and oral bacterial total counts, through a randomized controlled clinical trial with dogs. Furthermore haematological, and biochemical parameters as well as general side effects were monitored during the trial.

The study covers some issues that have been overlooked in other similar topics. The structure of the manuscript appears adequate and well divided in the sections. Moreover, the study is easy to follow, but some issues should be improved. Some of the comments that would improve the overall quality of the study are:

a. Authors must pay attention to the technical terms acronyms they used in the text.

b. Better stated the limitation of the study.

c. Conclusion Section: This paragraph required a general revision to eliminate redundant sentences and to add some "take-home message".

Author Response

Dear Reviewer,

We would like to thank you for the comments and the discussion points presented.

A complete revision of the manuscript was performed to avoid missed terms and duplications. The text referring to the limitations of the study and the conclusions section were also revised. The limitations of the study were further discussed (see lines 85-96). The discussion section was reorganized in order to avoid duplication of the information presented in the conclusion section.

All changes performed were highlighted in yellow in the new document.

Best Regards,

Eva Cunha

Reviewer 6 Report

Dear authors,

After the review process, I have several comments: you should clearly mention the aim of the paper in the abstract, not the literature data; numerical data also should be included in the abstract; you should include new perspectives for the study started from the utilization of bioactive compounds from natural products/sources, separation, characterization and possible new applications should be included; statistical data should be added to tables.

Best regards!

Author Response

Dear Reviewer,

We appreciate all your comments. Considering the abstract, we performed some modifications in accordance with your comments and the goal of the project is presented in the abstract, lines 15-18. Also, in the discussion section we included a paragraph about new perspectives and other bioactive products in research for PD control (see lines 97-107). The statistical data was added in tables 5 and 6.

Best Regards,

Eva Cunha

Round 2

Reviewer 3 Report

The Authors responded to all the comments of the Reviewers and improved the manuscript considerably. I have no further comments.

Reviewer 4 Report

The importance of anaerobic bacteria is well known even in the veterinary oral microbiology field (not only in the human). The referee regrets to mention here that the authors should have consulted with oral microbiologists before starting their projects.

Reviewer 6 Report

Dear authors,

no other comments rather than the first review.

Best regards!